# Increased Inflammation and Cardiometabolic Risk in Individuals with Low *AMY1* Copy Numbers

**DOI:** 10.3390/jcm8030382

**Published:** 2019-03-19

**Authors:** Clara Marquina, Aya Mousa, Regina Belski, Harry Banaharis, Negar Naderpoor, Barbora de Courten

**Affiliations:** 1Monash Centre for Health Research and Implementation, School of Public Health and Preventive Medicine, Monash University, Melbourne 3004, Australia; clara.marquina@monash.edu (C.M.); aya.mousa@monash.edu (A.M.); negar.naderpoor@monash.edu (N.N.); 2Faculty of Health, Arts and Design, Swinburne University of Technology, Melbourne 3122, Australia; rbelski@swin.edu.au; 3Department of Medicine, University of Melbourne, Melbourne 3010, Australia; hbanaharis@me.com

**Keywords:** amylase, *AMY1* copy numbers, salivary amylase gene, obesity, cardiometabolic risk, insulin sensitivity, insulin secretion, inflammation

## Abstract

Lower copy number variations (CNVs) in the salivary amylase gene (*AMY1*) have been associated with obesity and insulin resistance; however, the relationship between *AMY1* and cardiometabolic risk has not been fully elucidated. Using gold-standard measures, we aimed to examine whether *AMY1* CNVs are associated with cardiometabolic risk factors in an overweight or obese, otherwise healthy population. Fifty-seven adults (58% male) aged 31.17 ± 8.44 years with a body mass index (BMI) ≥25 kg/m^2^ were included in the study. We measured *AMY1* CNVs (qPCR); anthropometry (BMI; body composition by dual-energy X-ray absorptiometry); cardiovascular parameters (blood pressure, serum lipids by ELISA); insulin sensitivity (hyperinsulinaemic–euglycaemic clamp), insulin secretion (intravenous glucose tolerance test), and serum inflammation markers (multiplex assays). Based on previous studies and median values, participants were divided into low (≤4) and high (>4) *AMY1* CNV groups. Low *AMY1* carriers (*n* = 29) had a higher fat mass (40.76 ± 12.11 versus 33.33 ± 8.50 kg, *p* = 0.009) and LDL-cholesterol (3.27 ± 0.80 versus 2.87 ± 0.69 mmol/L, *p* = 0.038), and higher serum levels of interleukin [IL]-6, IL-1β, tumour necrosis factor-alpha and monocyte chemoattractant protein-1 (MCP-1) (all *p* < 0.05) compared with high *AMY1* carriers (*n* = 28), but there were no differences in glycaemic measures, including insulin sensitivity or secretion (all *p* > 0.1). Except for MCP-1, the results remained significant in multivariable models adjusted for age, sex, and fat mass (all *p* < 0.05). Our findings suggest that low *AMY1* CNVs are associated with increased cardiovascular disease risk and inflammation, but not glucose metabolism, in overweight or obese adults.

## 1. Introduction

Obesity is a multifactorial disorder that leads to an increased risk of cardiometabolic diseases including type 2 diabetes mellitus (T2DM) and cardiovascular diseases. The prevalence of obesity has tripled since 1975, and is expected to continue increasing [1]. Therefore, gaining a better understanding of the underlying risk factors of obesity is crucial for developing effective prevention and management strategies. It is well recognised, based on family and twin studies, that body mass index (BMI) is a highly heritable trait, with between 40–70% of its variance being attributable to genetic factors [2]. However, the common genetic variants identified in genome-wide association studies account for only a small proportion of population variability [2]. Recently, copy number variations (CNVs) in the salivary ⍺-amylase gene *AMY1* have emerged as potential contributors to this heritability.

Alpha-amylases catalyse the hydrolysis of starch into disaccharides and trisaccharides by catalysing the hydrolysis of α-1,4 glycosidic bonds [3]. In humans, there are two isoforms: salivary ⍺-amylase, the most abundant protein in the saliva, encoded by the genes *AMY1A*, *AMY1B*, and *AMY1C*, and pancreatic ⍺-amylase, encoded by *AMY2A* and *AMY2B* [4]. Serum α-amylase consists of both isoforms in a 1:1 proportion. Besides the digestive tract, salivary α-amylase is also expressed in several other tissues such as the central nervous system, the liver, uterus, mammary tissue, and testes [5]. CNVs of the *AMY1* gene have been reported to be associated with the levels and activity of ⍺-amylase in serum and saliva [6,7,8].

The first genetic link between obesity and *AMY1* came from a longitudinal study by Falchi et al. [9], which described an association between low CNVs of the *AMY1* gene and a higher risk of obesity measured by BMI. However, subsequent observational studies have produced inconsistent results. Some studies found no association between *AMY1* CNVs and obesity [10,11,12], while others reported that lower *AMY1* CNVs are associated with a higher BMI in prepubertal boys [13], in young male and female adults [14], and in a cohort of 597 Mexican children [15], as well as with early-onset obesity in a female Finnish population [16]. Conflicting results have also been reported regarding *AMY1* and glycaemic outcomes. A study of 1257 Korean men found that low *AMY1* CNVs were associated with increased insulin resistance measured by homeostatic model assessment of insulin resistance (HOMA-IR), but there was no relationship with BMI [17]. In a recent clinical trial of 692 overweight or obese adults receiving different dietary interventions, those with genetic *AMY1* variants that were associated with higher ⍺-amylase levels and activity had greater weight loss [18]. However, another study of an Asian cohort (*n* = 75) found no association between salivary ⍺-amylase activity or *AMY1* CNVs and postprandial glycaemic responses following the ingestion of a starch-rich meal (white rice) [19].

Despite a number of studies examining *AMY1* in relation to obesity and glucose metabolism, most studies have relied on indirect proxy measures such as BMI or HOMA-IR, and have not focused specifically on high-risk, overweight, or obese populations. To our knowledge, no studies have examined the relationship between *AMY1* and gold-standard measures of insulin sensitivity or insulin secretion, and few have used gold-standard adiposity measurements. Moreover, the relationship between *AMY1* and other markers of cardiometabolic risk, such as chronic low-grade inflammation, have not previously been explored.

We aimed to address these knowledge gaps by examining the relationship between *AMY1* CNVs and cardiometabolic risk factors using gold-standard measures of adiposity, insulin sensitivity, and insulin secretion, as well as with cardiovascular parameters and markers of chronic low-grade inflammation in a high-risk cohort of overweight or obese, otherwise healthy adults.

## 2. Materials and Methods

### 2.1. Study Design and Participants

This cross-sectional study utilises baseline data collected from participants who took part in a previous randomised trial investigating the role of vitamin D in the pathogenesis of T2DM [20]. For this study, data were available for 57 non-diabetic adults with overweight (BMI 25–29.9 kg/m^2^) or obesity (BMI ≥ 30 kg/m^2^), aged between 18–60 years old. On screening, participants underwent a detailed medical history and physical examination, routine blood analyses, anthropometric assessments, and a 75-g oral glucose tolerance test (OGTT) to exclude diabetes based on World Health Organization criteria [21]. All of the participants were characterised for body composition, glucose metabolism, and cardiovascular parameters, serum concentrations of inflammation markers, and *AMY1* copy numbers. Dietary habits were recorded using a validated questionnaire (three-day food record analysed on Foodworks 8.0 Professional; Xyris Software), which estimates the average amount of daily nutrient intake for starch, protein, saturated and total fat (all reported as g/day), and for total energy (kJ/day).

Participants were included in the study if they were overweight or obese, non-diabetic, non-smokers, not using illicit drugs or taking any medications or supplements at the time of the study, and had no clinical or laboratory signs of acute or chronic inflammation based on medical history or physical or laboratory examinations. Exclusion criteria included: age <18 or >60 years; high alcohol use (>4 and >2 standard drinks per week for males and females, respectively); current medical conditions or psychiatric disorders; active cancer within the preceding five years; and women who were pregnant and/or lactating or peri-menopausal or post-menopausal. This study received ethical approval from the Monash University Human Research Ethics Committee (CF13/3874-2013001988) and complied with the Declaration of Helsinki (2004) [22]. All of the participants provided written informed consent prior to study entry.

### 2.2. Anthropometric and Clinical Measures

BMI was calculated as weight in kilograms divided by height in meters squared. Body composition (total % body fat) was determined by dual-energy X-ray absorptiometry (DPX-L; Lunar Radiation, Madison, WI, USA). Then, total body fat was used to calculate fat mass (weight (kg) × total % body fat (decimal) = fat mass (kg)) as well as fat-free mass (weight (kg) − fat mass (kg) = fat-free mass (kg)).

Resting systolic and diastolic blood pressure (SBP/DBP) were assessed with an automated oscillometric measurement system (M6 Automatic BP monitor, Omron, Japan) following a 20-minute rest. Mean BP was calculated from the average BP from three different measurements.

A two-hour 75-g OGTT was performed following a 12-h overnight fast, and glucose tolerance status was determined by World Health Organisation (WHO) criteria [21]. Hyperinsulinaemic–euglycaemic clamps were performed to determine insulin sensitivity as described in detail elsewhere [20]. Briefly, an intravenous bolus injection of insulin (9 mU/kg) was administered, after which insulin was constantly infused (40 mU.m–2.min) for at least 120 min, and glucose was variably infused, monitored every 5 min, and adjusted until a steady state of euglycaemia was achieved (~5 mmol/L for the last 30 min). Insulin sensitivity (*M*-value) represents the weight-adjusted glucose infusion rate at which this steady state was achieved. As previously described [20], insulin secretion was measured using intravenous glucose tolerance tests whereby 50 mL of 50% glucose was delivered intravenously over a 3-min period, and the serum insulin area under the concentration–time curve (AUC) was determined using the trapezoidal rule [23]. Serum insulin AUC calculated 3 to 5 min after the glucose bolus was used to determine first-phase insulin secretion.

### 2.3. Biochemical Measures

All of the blood samples were analysed using standard quality control systems (all results within ± 2 SD) by accredited laboratories (Monash Health and Monash University Pathology). Plasma glucose concentrations were determined by the glucose oxidase method (YSI 2300 STAT Glucose & Lactate Analyser, YSI Inc., Yellow Springs, OH, USA). Serum insulin was measured by simultaneous immunoenzymatic sandwich assays (Access/DXI ultrasensitive insulin assay, Beckman Coulter, Australia), with inter-assay and intra-assay coefficients of variation (CVs) of <5% and <7%, respectively. Triglycerides, total cholesterol, low-density lipoprotein (LDL), and high-density lipoprotein (HDL) cholesterol in serum were determined using standard commercial enzymatic assays (LX20PRO Analyser and SYNCHRON Systems Lipid and Multi Calibrators, Beckman Coulter, Australia).

Plasma high-sensitivity C-reactive protein (hsCRP) was measured using highly sensitive near-infrared particle immunoassay rate methodology on a Synchron LX system analyser (Beckman Coulter, Australia). Intra-assay and inter-assay CVs for hsCRP were <5%. Serum cytokines and adipokines, including tumour necrosis factor alpha (TNF-α), interleukin (IL)-6, IL-1β, monocyte chemoattractant protein-1 (MCP-1), leptin, and adiponectin were quantified simultaneously using bead-based multiplex assays (LEGENDplex™, Human inflammation and metabolic panels, BioLegend, San Diego, CA, USA) following the manufacturer’s instructions. Data acquisition was performed using a BD™ LSR II flow cytometer and FACS DIVA software (Becton Dickinson, San Diego, CA, USA) and analysed using the LEGENDplex™ data analysis software (BioLegend, San Diego, CA, USA) with standard curves generated from 0 to 50,000 pg/mL for inflammatory markers and 0–200 ng/mL for adipokines, and samples were adjusted for dilution factors. Intra-assay and inter-assay CVs for all analytes were <9%.

### 2.4. Genotyping

*AMY1* copy numbers were measured in biobanked samples of peripheral blood mononuclear cells (PBMCs). PBMCs were isolated by centrifuging fasting whole blood samples collected in BD Vacutainer^®^ CPT™ cell preparation tubes with sodium citrate. The harvested PBMC pellet was resuspended in fetal bovine serum (FBS) with 10% dimethylsulphoxide (DMSO) and stored at −80 °C until analysis. Genomic DNA from PBMCs was isolated using the Maxwell 16 Cell LEV DNA purification kit (Promega, Fitchburg, WI, USA), and its purity and concentration was assessed using a NanoDrop One spectrophotometer (Thermofisher, Waltham, MA, USA). *AMY1* gene copy numbers were estimated by duplex quantitative real-time PCR (2qPCR) on a Life Technologies QuantStudio™ 12K Real-Time PCR system, with QuantStudio™ software version 1.2.2, with a protocol adapted from Falchi et al. [9], and analysed with CopyCallerR software version 2.1 (Thermo Fisher Scientific, Waltham, MA, USA). Each sample reaction consisted of two TaqMan CNV assays (Life Technologies, Carlsbad, CA, USA); one specific for the target, *AMY1* (Hs07226362_cn), and one specific for the reference gene (RNase P). Each sample was run in quadruplicate, and each run included an externally validated 14-copy control (NA18972, Coriell Cell Repositories, Camden, NJ, USA) along with a negative control.

### 2.5. Statistical Analyses

This is a secondary analysis of baseline data from participants who took part in a previous randomised trial, and the power calculation was based on the primary outcome of insulin sensitivity in the main trial [20]. Sample characteristics are presented as mean ± SD or frequencies (%) for continuous and categorical variables, respectively. The normality of variables was inspected visually using histograms and scatterplots, and continuous variables were logarithmically transformed to the base 10 if normality was violated. Differences between groups were examined using independent Student’s *t*-tests for continuous variables and chi-square tests for categorical variables. Predetermined variables known to influence cardiometabolic risk, such as age, sex, and fat mass, were included as covariates in multivariable linear regression models. In exploratory analyses, the influence of dietary factors such as starch consumption and carbohydrate content were included as continuous variables in additional multivariable models. Statistical analyses were performed using JMP V.14.1 (SAS Institute Inc., Cary, NC, USA) and Stata V.15.0 (StatCorp LP, College Station, TX, USA) statistical software. All of the tests were two-sided and *p*-values < 0.05 were considered statistically significant. 

## 3. Results

### 3.1. Sample Characteristics

Fifty-seven participants (34 male and 23 female) aged 31.17 ± 8.44 years (mean ± SD) were included in the study. The sample comprised 28% Caucasian, 39% South and Central Asian, and 23% Northeast/Southeast Asian ethnic groups, while 6% belonged to other ethnicities (African, Middle Eastern, Polynesian, and South American). Participants had a mean BMI of 31.5 ± 4.6 kg/m^2^ and a mean total percentage body fat of 41.1 ± 8.4%. The number of *AMY1* copies in the sample ranged from one to 13, with a median of four, which was used to define the low *AMY1* CNV group (≤4 copies) and the high *AMY1* CNV group (>4 copies) in accordance with previous studies [9]. The demographic, clinical, and biochemical characteristics of the participants within each group are presented in Table 1. There were no differences between the low and high *AMY1* CNV groups regarding demographic characteristics (Table 1) or dietary intakes of key macronutrients, including starch consumption and carbohydrate content as well as protein, fat, and total energy intakes (Appendix A).

### 3.2. Differences in Anthropometric Parameters between AMY1 CNV Groups

Compared with participants in the high *AMY1* CNV group, the low *AMY1* CNV group had significantly higher fat mass, as well as a trend towards higher BMI (Table 1). There were no differences in total body fat or fat-free mass between groups (both *p* ≥ 0.1). In multivariable models adjusted for age and sex, differences in fat mass remained significant (*p* = 0.016, Table 2), while results for other anthropometric variables were not altered (all *p* > 0.05; Table 2). In exploratory analyses, additional adjustment for dietary factors including starch, protein, total and saturated fat, total energy, and carbohydrate consumption did not alter any of these results (data not shown).

### 3.3. Differences in Glycaemic Parameters between AMY1 CNV Groups

There were no significant differences in insulin sensitivity between low and high *AMY1* CNV groups (*M*-value= 6.3 ± 2.9 and 6.5 ± 2.9 mg/kg/min, respectively, *p* = 0.7; Table 1). First-phase insulin AUC also did not differ significantly between groups (354.5 ± 250.9 mU/L for the low *AMY1* group and 419.4 ± 320.6 mU/L for the high *AMY1* CNV group, *p* = 0.7; Table 1). Results remained non-significant after adjusting for age and sex, as well as with additional adjustment for fat mass (Table 2). Additional adjustment for dietary factors did not alter results. There were no significant differences between low and high *AMY1* CNV groups for fasting and 2-h blood glucose, fasting insulin, or HOMA-IR in both univariable and multivariable analyses (all *p* > 0.05).

### 3.4. Differences in Cardiovascular Parameters between AMY1 CNV Groups

The low *AMY1* CNV group had a significantly higher concentration of serum LDL cholesterol compared with the high *AMY1* CNV group (*p* = 0.038; Table 1). Differences in LDL cholesterol remained significant after adjustment for age and sex and after additional adjustment for fat mass (*p* = 0.01 for both models, Table 2) and dietary factors (data not shown). There were no significant differences in other serum lipids, including total or HDL cholesterol or triglycerides before or after adjustment for covariates (all *p* > 0.05; Table 1 and Table 2). Other cardiovascular parameters including SBP and DBP did not differ between groups before or after adjustment.

### 3.5. Differences in Serum Inflammation Markers and Adipokines between AMY1 CNV Groups

Low *AMY1* copy number carriers had significantly higher concentrations of serum inflammation markers including IL-6 (*p* = 0.02), IL-1β (*p* = 0.01), TNF-α (*p* = 0.02), and MCP-1 (*p* = 0.03) compared with high *AMY1* copy number carriers (Table 1), but there were no differences in hsCRP (*p* = 0.5, Table 1). Differences in these markers between low and high *AMY1* groups remained significant after adjusting for age and sex (all *p* ≤ 0.04, Table 2), as well as after adjustment for fat mass (all *p* ≤ 0.01, Table 2), except for MCP-1, which was attenuated upon the inclusion of fat mass in the model (*p* = 0.06; Table 2).

Serum adiponectin and leptin concentrations were significantly higher in low *AMY1* CNV carriers compared to the high *AMY1* CNV group (*p* = 0.04 for both, Table 1). After adjustment for age and sex, differences in adiponectin remained significant (*p* = 0.04), and a trend persisted for differences in leptin (*p* = 0.05). After adjustment for fat mass, neither adiponectin nor leptin concentrations were different between groups (both *p* ≥ 0.1; Table 2). Additional adjustment for dietary factors in exploratory analyses did not alter results for any of the inflammation markers or adipokines.

## 4. Discussion

To our knowledge, this is the first study to examine the relationship between *AMY1* CNVs and gold-standard measures of insulin sensitivity and secretion, as well as serum inflammation markers in humans. We found that individuals with four or less copies of the *AMY1* locus had worse cardiometabolic profiles, including higher adiposity and LDL cholesterol concentrations, and more pronounced inflammation, compared with individuals with a higher number of *AMY1* copies. These differences persisted after adjustment for predetermined clinically relevant covariates. However, there were no differences in any of the glycaemic parameters between *AMY1* CNV groups. Our findings suggest that *AMY1* CNVs may be associated not only with obesity, but also with increased cardiovascular risk and chronic low-grade inflammation in overweight or obese individuals.

Our data showed that overweight or obese, but otherwise healthy adults with low *AMY1* copy numbers had significantly higher fat mass than individuals within the high copy number group. A trend was also observed for a higher BMI in low compared with high *AMY1* carriers. Our results are consistent with several cross-sectional studies that reported an inverse relationship between *AMY1* copy numbers and obesity [9,13,14,15]. A recent Mendelian randomisation analysis showed a bidirectional relationship between BMI and *AMY1* [8]. The authors described a significant, albeit limited, effect of higher *AMY1* CNVs on BMI and strong associations between *AMY1* or *AMY2* enzymatic activity and lower BMI [8]. However, other reports, including a large cross-sectional analysis of more than 3400 European individuals, did not find any associations between the number of copies of the *AMY1* gene and BMI [10]. Regarding other measures of adiposity, most parameters have not been extensively studied in relation to *AMY1* CNVs. A recent study that used bioelectric impedance, which is an indirect measurement of adiposity, found no significant association between obesity and *AMY1* CNVs [12]. Only two previous studies reported that increased fat mass, as measured by gold-standard dual energy X-ray absorptiometry (DEXA), was associated with low *AMY1* CNVs in Scandinavian populations [9,16], which is consistent with our findings. We add to the currently limited evidence by showing that, in a well-characterised multiethnic cohort, low *AMY1* carriers had higher fat mass measured by DEXA, compared with high *AMY1* carriers. 

Putative mechanisms by which *AMY1* may be linked to obesity include its role in taste perception and the digestion of starch. Increased *AMY1* copy numbers may be an evolutionary adaptation to high starch diets [6], and higher *AMY1* CNVs are proposed to improve the digestion of starchy foods as well as texture and taste perception, and therefore carbohydrate dietary habits [7]. Interestingly, differences in fat mass in our study remained significant after adjusting for starch and carbohydrate consumption, suggesting that *AMY1* CNVs may have other functions extending beyond its role in digestion. This is supported by a recent metabolomic study that analysed sera from low (*AMY1* copy number ≤4) and high *AMY1* carriers (*AMY1* copy number ≥8), and suggested that low *AMY1* carriers may have an increased β-oxidation of fatty acids and reduced cellular glucose uptake [24]. Based on current evidence, the functions of salivary α-amylase as well as the expression pattern of *AMY1* are not well understood, and further studies are needed to fully elucidate their role in the pathophysiological processes underlying obesity.

We also report a significantly higher LDL cholesterol concentration in low *AMY1* carriers compared with high *AMY1* carriers, which persisted after adjusting for age, sex, and fat mass, as well as in exploratory analyses adjusted for dietary factors. This finding contrasts with earlier reports [8,17] that found no associations between *AMY1* and LDL cholesterol in a population of more than 1000 healthy Korean males and in a large European cohort (*n* > 3400) of normal and overweight individuals. In fact, this is the first study to report a significant difference in LDL cholesterol between individuals with high and low *AMY1* CNVs. The higher BMI of our population may have enhanced the differences in serum lipid levels between low and high *AMY1* CNVs, hence allowing us to detect these differences. Overall, our finding of higher LDL cholesterol in low versus high *AMY1* carriers was robust and independent of multiple covariates, suggesting a potential link between *AMY1* CNVs and lipid metabolism. LDL cholesterol is an important risk factor for cardiovascular diseases; thus, further studies are needed to determine the effect of *AMY1* CNVs on the development of cardiovascular diseases and clarify the underlying mechanisms.

Using gold-standard methodology to measure insulin sensitivity and secretion, we found no differences in these parameters between the low and high *AMY1* groups. Previous reports of the association between *AMY1* and glycaemic outcomes have been inconsistent. *AMY1* CNVs did not affect glycaemic responses to a starch meal in 75 healthy Asian males [19]. Yet, another study comparing dietary interventions with different macronutrient composition reported a significant difference in HOMA-IR based on *AMY1* CNVs [18]. Similarly, in a cross-sectional study of 1257 healthy Korean men, Choi et al. [17] reported that low *AMY1* CNVs were associated with higher insulin resistance measured by HOMA-IR. However, in a small interventional trial (*n* = 14) [25], individuals with higher salivary α-amylase activity had lower postprandial glycaemic responses following starch ingestion. Importantly, salivary α-amylase levels in serum or saliva may not be fully explained by *AMY1* CNVs [9], and can be influenced by stress, metabolic status, or starch consumption. Discrepancies in the evidence may also be due to the different populations and methods used across studies. Our study is the first to employ gold-standard hyperinsulinaemic euglycaemic clamps and intravenous glucose tolerance tests, whereas previous studies have relied on surrogate measures of insulin sensitivity and secretion, such as HOMA-IR, HOMA-β, and the Matsuda index. Moreover, insulin sensitivity has been reported to be associated with ethnicity in adult populations [26]. The multiethnic character of our population, coupled with the relatively small sample size of our study, may have affected our ability to detect differences in these outcomes. Larger studies using similar gold-standard methodologies and exploring ethnic variation are needed to confirm our results.

Finally, another novel finding of our study is that, compared with high *AMY1* CNV carriers, individuals with low *AMY1* CNVs had greater low-grade chronic inflammation indicated by higher concentrations of serum cytokines, including IL-6, IL-1β, MCP-1, and TNF-α. Adipokines including leptin and adiponectin were also higher in low *AMY1* carriers. However, this was attenuated after adjusting for fat mass, suggesting that differences in adipokines were driven by differences in adiposity rather than *AMY1* CNVs per se. The association with serum cytokines, on the other hand, remained significant in all the models, suggesting a potentially important relationship between subclinical inflammation and *AMY1* CNVs that is independent of obesity and starch intake. Cytokines and adipokines are released not only by fat cells, but also by immune cells that infiltrate the tissue [27]. *AMY1* mRNA is expressed in adipose tissue [9], and it may affect their expression or release by adipocytes or immune cells. Moreover, IL-6 and MCP-1 have been suggested to regulate systemic glucose and/or lipid metabolism [27]; hence, *AMY1* CNVs could influence the levels of these markers by altering lipid metabolism, and their levels could be influenced by altering metabolism in a feedback loop. At this stage, explanations for this finding remain speculative, since no other study has explored the relationship between *AMY1* CNVs and subclinical inflammation.

Our study has some limitations. The cross-sectional design precludes causality, and potential confounding remains a possibility. This is a secondary analysis, and there was no formal power calculation; hence, the sample size may have been too small to detect differences in some parameters, including glycaemic measures. Per protocol, we recruited only overweight or obese individuals who were otherwise healthy; hence, our results may not be generalisable to other populations, including lean individuals or those with pre-existing conditions. Despite adjusting for age in multivariable analysis, participants in this study had a wide age range (18–57 years), and this may have influenced their cardiometabolic outcomes. *AMY1* CNVs are likely to be influenced by ethnicity; however, due to the small numbers of participants, we were unable to explore the influence of ethnic variation, and the multiethnic nature of the cohort may have reduced our ability to detect some differences between groups. *AMY1* copy number genotyping was performed using qPCR, which is reported to have lower sensitivity compared to other techniques such as digital droplet PCR [17], and these variations in methods may explain some of the inconsistencies between studies. Nevertheless, several studies using droplet digital PCR (ddPCR) have reported similar results to our study, whereby *AMY1* was associated with BMI [16] and fat mass [16]. Moreover, the results of both ddPCR and qPCR methods were shown to be equivalent when performed on blood samples [8,28]. Hence, we expect that any variations in sensitivity in the present study would be minimal, since qPCR was performed on PBMC samples. Despite these limitations, this is the first study to examine the relationship between *AMY1* CNVs and insulin sensitivity and secretion using gold-standard methods, and the first to explore relationships with inflammation markers and adipokines. We included a metabolically well-characterised cohort of high-risk overweight or obese individuals, where there was no confounding by medication use or disease status, and we were able to incorporate other confounders including adiposity and diet in our analyses. These factors have seldom been considered in previous studies.

## 5. Conclusions

In summary, our data suggests that overweight or obese individuals with low *AMY1* CNVs have greater adiposity, less favourable lipid profiles, and increased chronic low-grade inflammation compared with high *AMY1* carriers, but no differences in glucose metabolism as measured by gold-standard methods. These results suggest that low *AMY1* CNVs may predispose individuals to adiposity and cardiovascular disease risk. Further studies are required to establish causality and elucidate the potential use of *AMY1* genotyping for risk prediction and the targeted prevention of disorders underpinned by obesity and inflammation.

## Figures and Tables

**Table 1 jcm-08-00382-t001:** Demographic, clinical, and biochemical characteristics of participants within low and high *AMY1* CNV groups.

Characteristics/Outcome Measures	Low *AMY1* (*n* = 29)	High *AMY1* (*n* = 28)	*p*
Age (y)	30.8 ± 8.6	31.5 ± 8.5	0.8
BMI (kg/m^2^)	32.7 ± 5.4	30.3 ± 3.5	0.05
Total body fat (%)	42.8 ± 8.0	39.4 ± 8.7	0.1
Fat mass (kg)	40.8 ± 12.1	33.3 ± 8.5	**0.009**
Fat-free mass (kg)	54.3 ± 14.4	51.3 ± 11.0	0.5
Insulin sensitivity (M; mg/kg/min)	6.3 ± 2.9	6.5 ± 2.9	0.7
First phase insulin AUC (mU/L)	354.5 ± 250.9	419.4 ± 320.6	0.5
Total cholesterol (mmol/L)	5.1 ± 1.0	4.8 ± 0.8	0.2
HDL cholesterol (mmol/L)	1.2 ± 0.3	1.1 ± 0.2	0.3
LDL cholesterol (mmol/L)	3.3 ± 0.8	2.9 ± 0.7	**0.038**
Triglycerides (mmol/L)	1.5 ± 0.8	1.7 ± 1.0	0.3
hsCRP (mg/L)	3.9 ± 4.7	2.4 ± 2.1	0.5
IL-6 (pg/mL)	52.9 ± 55.7	24.3 ± 22.9	**0.02**
IL-1β (pg/mL)	34.4 ± 25.1	17.9 ± 13.4	**0.01**
TNF-α (pg/mL)	68.4 ± 12.8	32.61 ± 5.3	**0.02**
MCP-1 (pg/mL)	1169.2 ± 1058.2	613.9 ± 319.5	**0.03**
Adiponectin (ng/mL)	11906.3 ± 14777.5	7431.9 ± 10233.0	**0.04**
Leptin (ng/mL)	19.2 ± 32.7	11.1 ± 23.8	**0.04**

Data are expressed as mean ± SD. Low *AMY1* group is defined as ≤4 copies and high *AMY1* group as >4 copies. Differences between groups were analysed using independent Student’s *t*-tests and bold values indicate statistical significance at *p* < 0.05. Variables which did not fit a normal distribution were log-transformed to the base 10 to approximate normality prior to analysis. BMI, body mass index; AUC, area under the curve; HDL/LDL, high-density/low-density lipoprotein; hsCRP, high sensitivity C-reactive protein, IL, interleukin, TNF-α, tumour necrosis factor-alpha; MCP-1, monocyte chemoattractant protein-1.

**Table 2 jcm-08-00382-t002:** Multivariable linear regression models examining associations between cardiometabolic parameters and *AMY1* CNV groups after adjustment for covariates.

Dependent Variable	Model 1 (Adjusted for Age and Sex)				Model 2 (Adjusted for Age, Sex, and Fat Mass)			
	β (95% CI)	R^2^	*t*	*p*	β (95% CI)	R^2^	*t*	*p*
BMI (kg/m^2^)	−2.3 (−4.7, 0.2)	0.1	−1.9	0.07				
Body fat (%)	−2.1 (−5.1, 0.8)	0.6	−1.4	0.2				
Fat mass (kg)	−6.8 (−12.3, −1.3)	0.2	−2.5	**0.016**				
Fat−free mass (kg)	−0.03 (−0.1, 0.003)	0.6	−1.9	0.07				
Insulin sensitivity (M; mg/kg/min)	0.2 (−1.4, 1.7)	0.03	0.2	0.8	−0.8 (−2.2, 0.7)	0.3	−1.1	0.3
First-phase insulin AUC (mU/L)	0.08 (−0.2, 0.3)	0.02	0.7	0.5	0.2 (−0.1, 0.4)	0.1	1.2	0.2
Total cholesterol (mmol/L)	−0.4 (0.9, 0.04)	0.2	−1.8	0.07	−0.5 (−0.9, 0.03)	0.2	−1.9	0.07
HDL-C (mmol/L)	−0.02 (−0.1, 0.02)	0.1	−1.0	0.3	−0.02 (−0.1, 0.03)	0.1	−0.7	0.5
LDL-C (mmol/L)	−0.5 (−0.8, −0.1)	0.2	−2.5	**0.01**	−0.5 (−0.9, −0.1)	0.2	−2.6	**0.01**
Triglycerides (mmol/L)	0.03 (−0.08, 0.1)	0.2	0.6	0.5	0.04 (−0.1, 0.2)	0.2	0.6	0.5
hsCRP (mg/L)	−0.1 (−0.4, 0.2)	0.02	−0.7	0.5	−0.1 (−0.3, 0.2)	0.03	−0.4	0.7
IL-6 (pg/mL)	−0.3 (−0.5, −0.02)	0.1	−2.2	**0.03**	−0.3 (−0.6, −0.1)	0.2	−2.6	**0.01**
IL-1β (pg/mL)	−0.6 (−1.1, −1.0)	0.2	−2.5	**0.02**	−0.7 (−1.2, −0.22)	0.2	−2.9	**0.005**
TNF-α (pg/mL)	−0.3 (−0.5, −0.02)	0.1	−2.16	**0.04**	−0.3 (−0.6, −0.07)	0.2	−2.6	**0.01**
MCP-1 (pg/mL)	−0.2 (−0.3, −0.006)	0.1	−2.1	**0.04**	−0.2 (−0.4, 0.01)	0.1	−1.9	0.06
Adiponectin (ng/mL)	−0.3 (−0.6, −0.02)	0.1	−2.1	**0.04**	−0.3 (−0.6, 0.07)	0.1	−1.6	0.1
Leptin (ng/mL)	−0.3 (−0.5, 0.005)	0.3	−2.0	**0.05**	−0.1 (−0.4, 0.1)	0.4	−1.0	0.3

Reference group: Low *AMY1* group. Data are presented as unstandardised beta coefficients with 95% confidence intervals and the corresponding t-statistic, R-squared value, and p-value for each model. Variables that did not fit a normal distribution were log-transformed to the base 10 to approximate normality prior to analysis. Model 1: adjusted for age and sex; Model 2: adjusted for age, sex, and fat mass. BMI, body mass index; AUC, area under the curve; HDL-C/LDL-C, high-/low-density cholesterol; hsCRP, high-sensitivity C-reactive protein, IL, interleukin, TNF-α, tumour necrosis factor-alpha; MCP-1, monocyte chemoattractant protein-1.

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
