# Peer review of "Increased Inflammation and Cardiometabolic Risk in Individuals with Low AMY1 Copy Numbers"

_jcm, 2019, doi:10.3390/jcm8030382_

Round 1
Reviewer 1 Report
This is the first study to examine the relationship between AMY1 CNVs and f insulin sensitivity and secretion, serum inflammation markers. It was found that individuals with 4 or less copies of the AMY1 locus had worse cardiometabolic profiles, higher LDL cholesterol, adiponectin, leptin and inflammatory markers concentrations, compared with individuals with a higher number of AMY1 copies.Adipokines differences were attenuated after adjusting for fat mass. Other differences persisted after adjustment for clinically relevant covariates. There were no differences in any of the glycaemic parameters between AMY1 CNV groups. Despite cross-sectional design and small study sample the presented data bring new information about a role of amylase gene in many pathways with potential adverse effect of low AMY1 on cardiovascular and metabolic risk and eventually human life span. Acknowledging study limitations, the presented data look very perspective and intriguing. Minor correction: line 86, overweight should be characterised as BMI 25-29.9 kg/m2..
Author Response
Response to Reviewer 1
· Comment 1: Despite cross-sectional design and small study sample the presented data bring new information about a role of amylase gene in many pathways with potential adverse effect of low AMY1 on cardiovascular and metabolic risk and eventually human life span. Acknowledging study limitations, the presented data look very perspective and intriguing. Minor correction: line 86, overweight should be characterised as BMI 25-29.9 kg/m2.
· Response: We thank the reviewer for their time and effort in reviewing our manuscript and providing positive feedback. As suggested by the reviewer, we have now modified the BMI classification of overweight to be more correctly stated as 25-29.9 kg/m2 (page 2, line 87).
Reviewer 2 Report
In the present manuscript, Marquina et al examine the association between AMY1 CNVs and gold standard measures of glycemic control, as well as anthropometrics, blood lipids and concentrations of various cytokines and adipokines, in an ethnically diverse population of 57 overweight/obese healthy adults. The authors identified associations between low AMY1 CNVs and some deleterious blood lipid and inflammatory parameters, as well as higher fat mass. They found no evidence of an association between AMY1 CNVs and measures of glycemic regulation. They concluded that individuals with low AMY1 CNVs have a worse cardiometabolic profile than those with a higher AMY1 CNV.
This is a well written, interesting manuscript and some of the findings, especially with respect to inflammatory cytokines, are intriguing. My only question revolves around the adjustments conducted for dietary intake. The authors mention that starch intake and carbohydrate content of the diet did not differ between groups, but what about protein and fat? Saturated vs. PUFA or MUFA content of the diet? It would be worthwhile to mention these results, if available. If not, the authors should consider carrying out a more thorough analysis of the potential influence of diet on the observed associations.
Author Response
Response to Reviewer 2
· Comment 1: This is a well written, interesting manuscript and some of the findings, especially with respect to inflammatory cytokines, are intriguing. My only question revolves around the adjustments conducted for dietary intake. The authors mention that starch intake and carbohydrate content of the diet did not differ between groups, but what about protein and fat? Saturated vs. PUFA or MUFA content of the diet? It would be worthwhile to mention these results, if available. If not, the authors should consider carrying out a more thorough analysis of the potential influence of diet on the observed associations.
· Response: We thank the reviewer for their time and insightful feedback on our manuscript. The reviewer raises an important point regarding the fat and protein content of the diet and we agree that these may influence the studied parameters. As recommended by the reviewer, we have now rerun the analysis to check for differences in other macronutrient components of the diet besides starch and carbohydrate content, including protein, total and saturated fat, as well as total energy. No differences between AMY1 groups were identified for these parameters and adjustment for these factors did not significantly alter the results. Nevertheless, we appreciate the importance of considering diet in the present study, and we have now incorporated these findings in the results section of the manuscript (page 4, lines 186-189) as well as in a supplementary table showing the macronutrient intake of low versus high AMY1 groups (Table S1, Supplementary File).
Reviewer 3 Report
The authors of the manuscript entitled "Increased inflammation and cardiometabolic risk in individuals with low AMY1 copy numbers" show an association between CNVs of the AMY1 gene and several parameters of adiposity, lipid metabolism and inflammation in adults with overweight or obesity. It is an interesting study that adds new data to the role of AMY1 variants in obesity and its related comorbidities.
The manuscript is well written and references are recent and relevant.
I have the following comments regarding the content of the manuscript:
1. The width of the age range should be discussed at some point in the discussion, despite analyses being adjusted by this variable.
2. Authors should specify in which units/type of variable they used starch consumption and carbohydrate content to adjust analyses. I.e. whether they used continuous or categorical variables. (page 4, line 168).
3. The multiethnic character of the studied population should be discussed, pointing to potential differences found in prior studies conducted in different populations.
4. Do the authors have data on circulating glucose and insulin values, as well as HOMA-IR of the participants? Perhaps there are differences in these parameters, with glucose tolerance not being affected. This information would definitely provide a better image of the situation.
5. What do authors mean in page 8, lines 329-330, by stating that the "cross-sectional design precludes causality and reverse causation remains a possibility"? It seems that they indicate that inflammation or obesity could influence CNVs, which is of course impossible. Please rephrase or better explain the idea.
As for spelling and other formatting details:
6. The abbreviation of a gene, when referring to the gene itself, should be written in capital itallics letters, such as AMY1. Please revise the document to fix this at appropriate places.
7. Page 3, Line 113, the reference of the WHO criteria for glucose tolerance should be mentioned here, too. I believe it is number 22.
8. Results section: there is no need to repeat data from tables in the text throughout the different subsections.
Author Response
Response to Reviewer 3
· Comment 1: The width of the age range should be discussed at some point in the discussion, despite analyses being adjusted by this variable.
· Response: We thank the reviewer for taking the time to thoroughly review our manuscript and provide helpful feedback. We agree with the reviewer that the wide age range is an important consideration, particularly when evaluating cardiometabolic outcomes. As recommended, we have now highlighted this point in the discussion section of the manuscript (page 8, lines 344-348).
· Comment 2: Authors should specify in which units/type of variable they used starch consumption and carbohydrate content to adjust analyses. I.e. whether they used continuous or categorical variables (page 4, line 168).
· Response: Diet content was collected using 3-day food diaries and analysed on Foodworks V.8. to estimate daily nutrient intakes as continuous variables. These were calculated as kJ/day for total energy intake and g/day for the remaining macronutrients. We acknowledge that this was not well-described in the manuscript and, following the reviewer’s suggestion, we have now clarified this in the methods section of the manuscript (page 3, lines 94-96; page 4, lines 172-174). We have also added a supplementary table showing the macronutrient intakes with their respective units for each group (Table S1, Supplementary File).
· Comment 3: The multiethnic character of the studied population should be discussed, pointing to potential differences found in prior studies conducted in different populations.
· Response: The reviewer highlights a valid point. Our multiethnic study population is an important factor to consider and, indeed, some studies have reported that ethnicity can impact cardiometabolic outcomes (Faramus et al. 2002, The Journal of Nutrition) as well as having a potential influence on the relationship between AMY1 and cardiometabolic risk (Elder et al. 2018, Expert Rev Endocrinol Metab). We acknowledge that the multiethnic nature of our cohort may explain some of the discrepancies with previous studies and in response to the reviewer’s suggestion, we have now modified parts of the discussion section to consider the potential influence of ethnicity on our findings (page 8, lines 315-316 and 319-323; and page 9, line 346-349).
· Comment 4: Do the authors have data on circulating glucose and insulin values, as well as HOMA-IR of the participants? Perhaps there are differences in these parameters, with glucose tolerance not being affected. This information would definitely provide a better image of the situation.
· Response: Although glucose and insulin concentrations as well as HOMA-IR were measured as part of the original study, we had decided to focus on the novel aspects that the present study could offer, that is, the use of gold-standard hyperinsulinaemic-euglycaemic clamp measures. However, we agree with the reviewer that including these additional measures of glucose metabolism would provide a clearer overall picture of the relationships between AMY1 and glycaemic outcomes. We have therefore reanalysed our data and found no differences in fasting blood glucose, 2-hour blood glucose post-OGTT, fasting insulin, or HOMA-IR between AMY1 groups, and we have now modified the results section of the manuscript to reflect these findings (page 6, lines 217-220).
· Comment 5: What do authors mean in page 8, lines 329-330, by stating that the "cross-sectional design precludes causality and reverse causation remains a possibility"? It seems that they indicate that inflammation or obesity could influence CNVs, which is of course impossible. Please rephrase or better explain the idea.
· Response: The reviewer is correct in that reverse causation is not possible in relation to CNVs - this was an oversight on our behalf and has now been removed from the manuscript (page 8, line 339-340).
· Comment 6: The abbreviation of a gene, when referring to the gene itself, should be written in capital italics letters, such as AMY1. Please revise the document to fix this at appropriate places.
· Response: We have now reformatted all the gene names to appear in capital italics throughout the manuscript (see tracked changes).
· Comment 7: Page 3, Line 113, the reference of the WHO criteria for glucose tolerance should be mentioned here, too. I believe it is number 22.
· Response: As recommended by the reviewer, we have now added the reference to the WHO criteria (reference 21) to the specified section of the manuscript (page 3, line 115-116).
· Comment 8: Results section: there is no need to repeat data from tables in the text throughout the different subsections.
· Response: We thank the reviewer for raising this point. We had initially included these data in the text to make it easier for the reader to interpret the results without having to refer to the tables. However, we recognize that there was some unnecessary repetition of data and as suggested by the reviewer, we have now revised the results section of the manuscript to remove some of the repeated data where appropriate, leaving only the most relevant results (page 5, lines 203-205; page 6, lines 231-233).
Round 2
Reviewer 2 Report
I would like to thank the authors for presenting the dietary intake data that I requested in my previous review. I also think the mention of ethnicity at various points throughout the manuscript makes it more thorough. I consider the manuscript worthy of publication in its present form and have no further suggestions.
Reviewer 3 Report
Authors have adequately fulfilled the prior comments and the manuscript has improved its quality after the review.